# Mode-Selective Photonic Lanterns for Orbital Angular Momentum Mode Division Multiplexing

**Yan Li [1,\*], Yang Li [1] , Lipeng Feng [1], Chen Yang [2], Wei Li [1], Jifang Qiu [1], Xiaobin Hong [1], Yong Zuo [1], Hongxiang Guo [1], Weijun Tong [2] and Jian Wu [1,\*]**

1   State Key Laboratory of Information Photonics and Optical Communications, Beijing University of Posts and Telecommunications, Beijing 100876, China; aslan_ly@163.com (Y.L.); 15201017279@163.com (L.F.); w_li@bupt.edu.cn (W.L.); jifangqiu@bupt.edu.cn (J.Q.); xbhong@bupt.edu.cn (X.H.); yong_zuo@bupt.edu.cn (Y.Z.); hxguo@bupt.edu.cn (H.G.)
2   State Key Laboratory of Optical Fibre and Cable Manufacture Technology, Yangtze Optical Fibre and Cable Joint Stock Limited Company, Wuhan 430074, China; chenyang@yofc.com (C.Y.); tongweijun@yofc.com (W.T.)
\*   Correspondence: liyan1980@bupt.edu.cn (Y.L.); jianwu@bupt.edu.cn (J.W.)

**Abstract:** We analyze the mode evolution in mode-selective photonic lanterns with respect to taper lengths, affected by possible mode phase differences varying along the taper. As a result, we design a three-mode orbital angular momentum (OAM) mode-selective photonic lantern by optimizing the taper length with mode crosstalk below −24 dB, which employs only one single mode fiber port to selectively generate one OAM mode.

**Keywords:** mode division multiplexing; orbital angular momentum; photonic lantern

## 1. Introduction

Mode division multiplexing (MDM) has attracted great interest due to its potential for addressing the forthcoming capacity crunch [1]. The orbital angular momentum (OAM) modes can be extensively used as separate channels in MDM communication networks due to the orthogonality [2]. As the key techniques, the generation and multiplexing of OAM modes in MDM systems have been achieved in free space [3], silicon photonics [4], and fiber-based configurations [5]. Due to the high insertion loss, high fabrication cost, and limited scalability, the free space and silicon photonics-based solutions are still subject to further improvements and upgrading. The fiber-based solutions play a promising role to perform OAM manipulations because of their compatibility with transmission fibers and compactness. They can be widely applied in practical communication systems. Some fiber-based solutions for OAM mode generation have been carried out [5–7], but a simple and feasible fiber-based OAM multiplexing method is still yet to be developed.

Meanwhile, all-fiber-based techniques, e.g., photonic lanterns (PLs), which offer the potential for direct integration with existing telecom/datacom infrastructures and low insert loss, are highly desirable [8]. A PL can be nearly lossless, scaled to many modes, and robust because it can be directly spliced to one few-mode fiber (FMF) and several single-mode fibers (SMFs) with the right design [9]. Previous work [9] demonstrated a $3 \times 1$ fiber-based photonic lantern spatial multiplexer with mode selectivity greater than 6 dB and transmission loss of less than 0.3 dB. PLs for linear polarized (LP) mode multiplexing have been reported, which can support 10 LP modes so far [10].

Recently, OAM mode division multiplexing in the systems containing PLs has been reported [11,12]. In [11], an OAM mode multiplexer using an annular core mode-selective photonic lantern (MSPL) is proposed. However, it cannot simultaneously multiplex both $OAM_{-1}$ and $OAM_{+1}$, as it has to employ

two input SMF ports of a MSPL to generate one OAM mode. We have previously proposed a one-port excited all-fiber OAM multiplexer based on cascading a MSPL and a mode-polarization controller [12], but the mode-polarization controller introduces instability and additional complexity.

In this paper, we analyze the periodic evolution of the PL output modes with respect to taper lengths, affected by possible mode phase differences varying along the taper. Then, we design a three-mode orbital angular momentum mode-selective photonic lantern (OAM-MSPL) utilizing the phase difference caused by the mode degeneracy breaking along a specific area of the taper. The $OAM_0$, $OAM_{-1}$, and $OAM_{+1}$ can be excited respectively when selectively and separately injecting light into the three SMF cores. Simulation results show that the mode crosstalk of the OAM-MSPL is below $-24$ dB.

## 2. Principle

A taper transition can couple light between one FMF and several SMFs. If the number of SMFs matches the number of spatial modes in the FMF, the transition can have low loss in both directions [13]. Taking a three-mode PL as an example, a standard PL is fabricated by adiabatically tapering three separate SMFs in a low-index glass jacket, as shown in Figure 1a. As unitary coupling between SMFs and one FMF is only possible by optimizing the arrangements of the cores [14], the cross section of the untapered end of the PL is shown in Figure 1b, which can be considered approximately as three isolated fiber cores (Fiber 1, 2, and 3 cores) with a circular cladding and a low-index jacket. The structure is tapered such that the SMF cores nearly vanish, the SMF cladding becomes the new few-mode core, and the low-index capillary becomes the FMF cladding. The tapered end of the PL matches the common two-mode fiber supporting $LP_{01}$ and $LP_{11}$ modes. The original $LP_{01}$ modes in the SMF cores eventually evolve into the $LP_{01}$ and $LP_{11}$ modes in the FMF core.

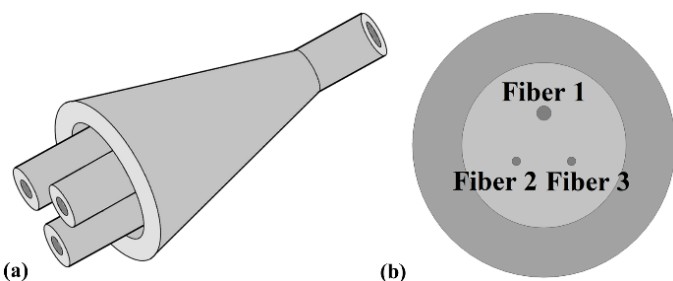

**Figure 1.** The (**a**) structure and (**b**) cross section of a photonic lantern (PL). Note: (**b**) is not to scale.

In this paper, we obtain the eigenmode profiles and effective refractive indexes of the PL at different positions along the taper, simulated by mode analysis using the finite element method (FEM). However, FEM is not able to simulate the many elements of the light transmission in the PL, such as the adiabatic taper criterion and the actual forms of the excited supermodes, i.e., local modes. It is not able to analyze the possible effects brought by the variation of the taper lengths either. Therefore, as a supplement, we use the beam propagation method (BPM) to analyze the evolution of the local modes excited by selectively injecting light into the SMF cores during the taper of the PL, in order to consider the variation of the light transmission conditions.

Three-mode MSPLs for LP mode multiplexing can be divided into two types: the standard three-mode MSPL designed with three totally different SMF cores, and the three-mode mode-group-selective PL owning a pair of symmetric SMF cores with a lower initially designed normalized frequency value (V value) compared to the other core. We simulate both these types to analyze the evolution of the eigenmodes and the local modes using FEM and BPM. In this paper, the simulated PL has three cores arranged in an equilateral triangle shape with a core pitch of 42.0 μm, as the reduced cladding fibers can be used to assist adiabaticity [15,16]. The core index is 1.4482. For the standard MSPL, the core diameters are 11.0 μm, 8.65 μm, and 6.55 μm, respectively. For the mode-group-selective PL, the core diameters are 11.0 μm, 6.55 μm, and 6.55 μm, as the Fiber 2 core and

the Fiber 3 core are identical. The cladding and jacket diameters are 125 and 1116 μm, and the cladding and jacket indexes are 1.444 and 1.4398, respectively.

As shown in Figure 2a, for the standard MSPL, the original eigenmodes of the untapered end are the $LP_{01}$ modes in the three separate SMF cores. The second- and third-highest indexes remain separate until the SMF cores nearly vanish and the local modes evolve into the degenerate $LP_{11}$ modes. When selectively injecting light into the Fiber 2 or 3 core, as shown in Figure 2b, at the same taper ratio, the local mode shown as the BPM profile is in correspondence with the eigenmode, shown as the FEM profile. On account of the separation of the effective refractive indexes, the second- and third-order local modes individually evolves into the two second-order eigenmodes in the FMF end, i.e., $LP_{11a}$ and $LP_{11b}$.

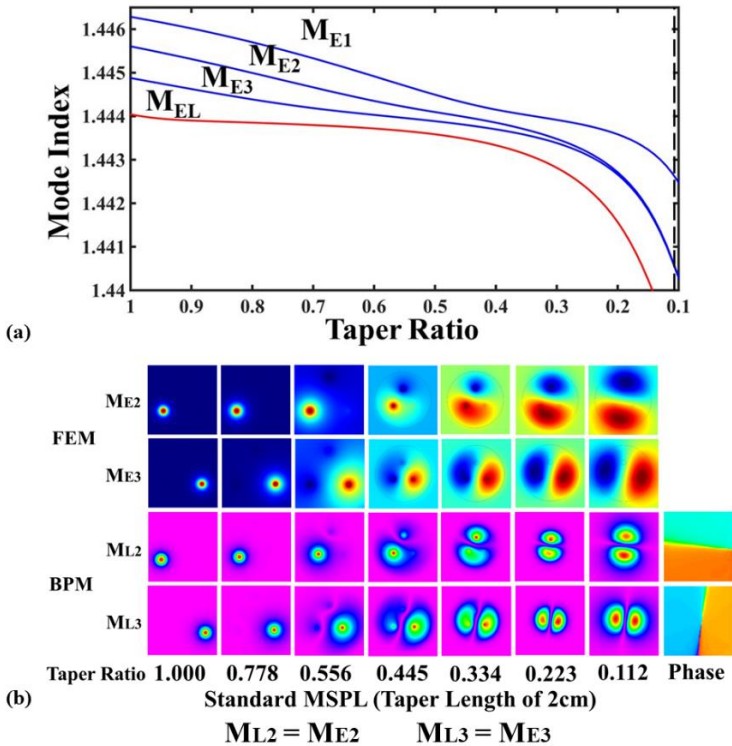

**Figure 2.** (**a**) The FEM simulation mode indexes evolution along the taper of the standard MSPL. $M_{E1}$, $M_{E2}$, and $M_{E3}$ represent the first- and the two second-order eigenmodes. The $M_{EL}$ represents the leaky mode simulated by FEM. The black dashed line indicates the taper ratio of 0.112. (**b**) The comparison of the FEM/BPM simulated mode profiles at different positions along the taper when selectively injecting light into the Fiber 2 and 3 cores of the standard MSPL. $M_{L1}$, $M_{L2}$, and $M_{L3}$ represent the local modes when individually injecting light into Fiber 1, 2, or 3 cores, respectively. BPM: beam propagation method; FEM: finite element method; MSPL: mode-selective photonic lantern.

As shown in Figure 3a, for the mode-group-selective PL, as the Fiber 2 and 3 cores share the same V value, the initial eigenmodes are the in-phase and reverse-phase combination of the $LP_{01}$ modes in the Fiber 2 and 3 cores, and their effective refractive indexes are identical at the starting area of the taper. When selectively injecting light into either of the pair of the totally symmetric and identical cores (Fiber 2 or 3 cores), both the degenerate second-order eigenmodes are excited. The two eigenmodes break the degeneracy and bring the difference value of the effective refractive indices in the deciding area of the taper (where the effective index differences of $M_{L2}$ and $M_{L3}$ are larger than $10^{-4}$, the value normally considered as the floor in order to eliminate MIMO-DSP in PLs [17]), then they return to degeneracy again at the end of the taper. As shown in Figure 3b, each of the local modes shown as the BPM profile is the isometric superposition of the two eigenmodes shown as FEM profiles, and the composition of the output mode in the FMF end depends on the accumulation of the phase differences

brought by the two eigenmodes in the deciding area of the taper. The relationship can be described as follows:

$$M_{L2} = M_{E2} + \exp(i\varphi)M_{E3} \tag{1}$$

$$M_{L3} = M_{E2} + \exp[i(\pi-\varphi)]M_{E3}, \tag{2}$$

where $M_{L2}$ and $M_{L3}$ represent the local modes when individually injecting light into the Fiber 2 or 3 core, $M_{E2}$ and $M_{E3}$ represent the two second-order eigenmodes, and $\varphi$ represents the final phase difference between the two parts of the excited eigenmodes generated during the taper. It is already known that one pair of degenerated $LP_{11}$ modes which satisfy the $\pi/2$ phase difference can be used to generate $OAM_{\pm 1}$ modes in optical fibers [7], as follows:

$$OAM_{-1} = LP_{11a} + i\,LP_{11b} \tag{3}$$

$$OAM_{+1} = LP_{11a} - i\,LP_{11b}. \tag{4}$$

Therefore, when selectively injecting light into the Fiber 2 or 3 core, if $\varphi$ equals $\pi/2$ (or $-\pi/2$), the output mode is one of the $OAM_{\pm 1}$ modes, and if $\varphi$ equals 0 (or $\pi$), the output mode is one of the $LP_{11}$ modes.

The BPM simulation mode profiles of the PLs with lengths of 2 cm, 2.65 cm, and 3.3 cm are shown in Figure 3b–d, as the final taper ratio is determined as 0.112. As described above, when injecting light into either the Fiber 2 or 3 core selectively, the variation of the phase differences leads to the changing of the superposed output mode. For instance, in the simulated PL with a taper length of 2 cm, the excited mode of the Fiber 2 core corresponds to $OAM_{-1}$. It becomes $LP_{11a}$ in the simulated PL with a taper length of 2.65 cm, and then turns to $OAM_{+1}$ when the taper length is 3.3 cm. The output mode profiles of the PLs with different taper lengths are shown in Figure 4, based on the BPM simulation when injecting light into each of the three SMF cores.

It has been shown that there is a periodic variation of the generating $OAM_{\pm 1}$ modes at a half cycle of 1.3 cm as the taper length increases. Hence, we can obtain a designed OAM-MSPL if the $LP_{11}$ modes satisfies the $\pi/2$ phase difference. Moreover, we could also optimize taper ratios and V values of the SMF cores to control the phase difference variation in order to obtain low-loss OAM-MSPLs.

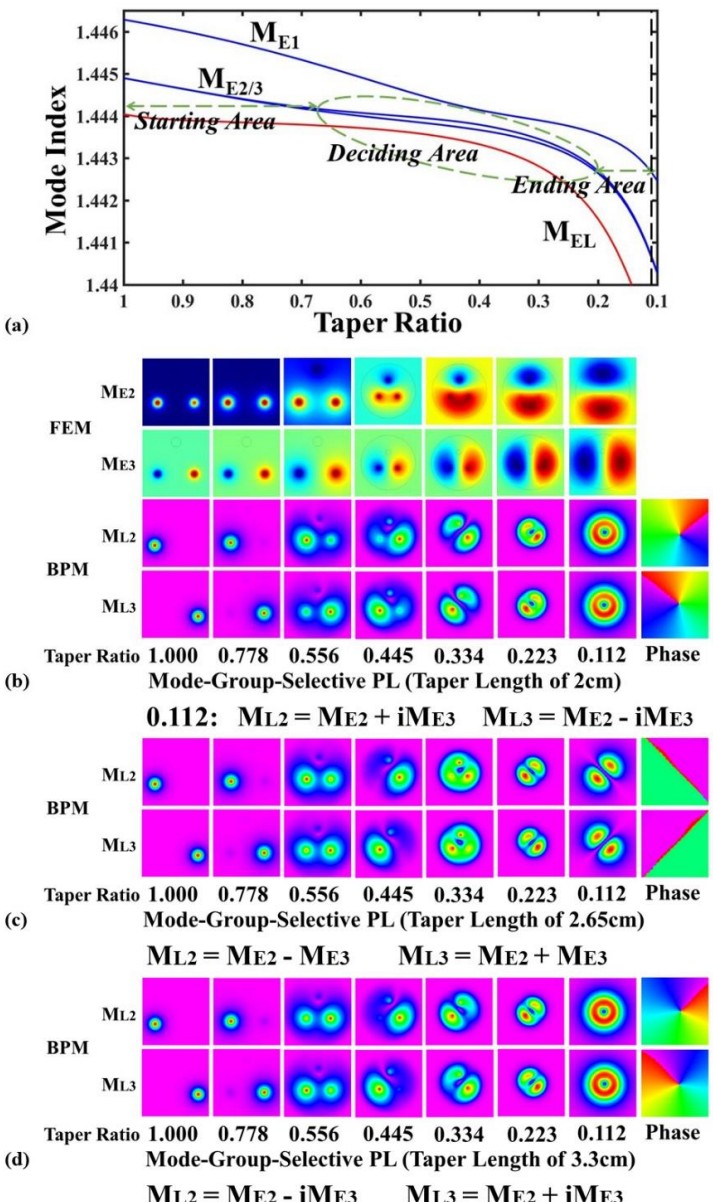

**Figure 3.** The FEM simulation mode indexes' evolution along the taper of the mode-group-selective PL (**a**). The comparison of the FEM/BPM simulated mode profiles at different positions along the taper when selectively injecting light into the Fiber 2 and 3 cores of the mode-group-selective PL with the taper length of (**b**) 2, (**c**) 2.65, and (**d**) 3.3 cm.

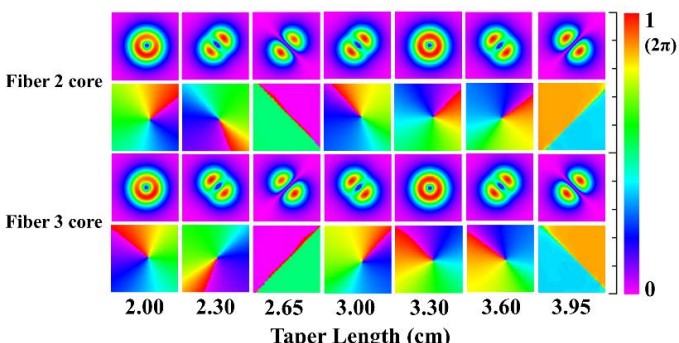

**Figure 4.** The BPM simulation output mode profiles of the PLs with different taper lengths when selectively injecting light into the Fiber 2 and 3 cores.

## 3. Simulation Results

As a result of the mode evolution analysis in MSPLs described above, we design the OAM-MSPL with a taper length of 2.0 cm, similar to practical PLs. When injecting light into the Fiber 1 core, the mode coupling efficiencies of $OAM_0$, $OAM_{-1}$, and $OAM_{+1}$ are 99.372%, 0.306%, and 0.306%, respectively, which were obtained by calculating the correlation coefficients between the target modes of the PL and the matching FMF as the waist [18]. As for Fiber 2, they are 0.318%, 99.665%, and 0.001%, respectively. As for Fiber 3, they are 0.318%, 0.001%, and 99.665%, respectively. Simulation results show good characteristics of multiplexing when the few-mode ends of one pair of the designed PLs are connected, as shown in Figure 5a.

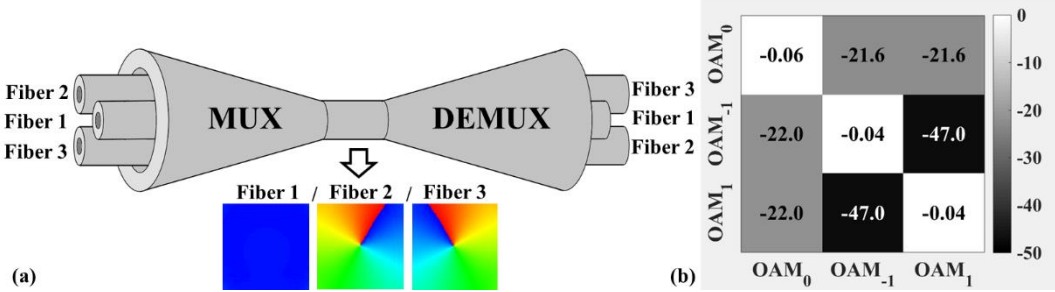

**Figure 5.** (**a**) The schematic of the multiplexer/de-multiplexer (MUX/DEMUX) simulation. (**b**) Calculated mode crosstalk matrix of one pair of the OAM-MSPLs during selective excitation (bottom axis) and measurement (left axis). The diagonal matrix elements represent the simulated insertion loss of one pair of the OAM-MSPLs. OAM-MSPLs: orbital angular momentum mode-selective photonic lanterns.

When injecting light into the target fiber core of the MUX-PL (where light is injected into the SMF ends and is emitted from the FMF end), we obtain the mode crosstalk, which is defined as the ratio of power in the corresponding fiber core of the DEMUX-PL (where light is injected into the FMF end and is emitted from the SMF ends) over power in the other cores. The matrix elements shown in Figure 5b are displayed in units of decibels (dB), with the mode crosstalk of one pair of the designed PLs below $-21$ dB, based on the symmetric MUX/DEMUX simulation structure. The individual simulation shows the mode crosstalk of the single PL is below $-24$ dB, which matches the result of the MUX/DEMUX simulation. The diagonal elements represent the simulated insertion loss (IL) of one pair of the designed PLs, which is below $-0.06$ dB, as the IL of a single PL is below $-0.03$ dB on average.

We calculate correlation coefficients between the target OAM modes and FMF via the overlap integral method by scanning the linear taper lengths and the light wavelengths, as shown in Figure 6a,d. With the mode crosstalk shown in Figure 6b,c, it is indicated that the OAM-MSPL has a taper length error tolerance longer than 1 mm, which can be controlled by common PL-tapering processors. The mode crosstalk of a single PL can be controlled below $-21$ dB, with the IL below $-0.06$ dB. Furthermore, as simulations performed, the designed PL works stably in the whole C-band and L-band with low loss and crosstalk.

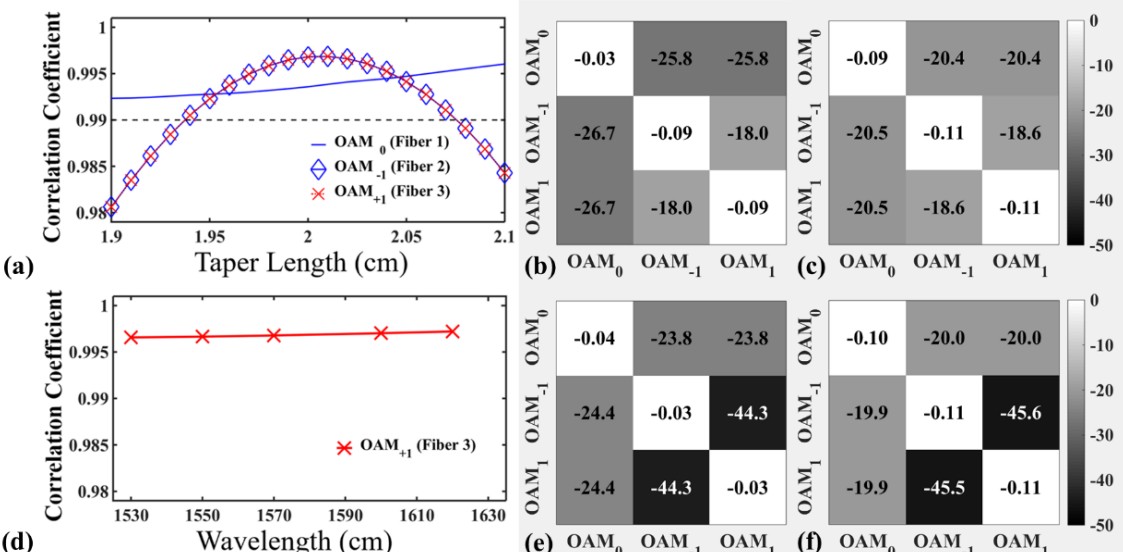

**Figure 6.** Calculated correlation coefficients of OAM-MSPL with (**a**) taper lengths around 2 cm when injecting light into all of the cores and (**d**) with wavelengths in the C-band and L-band when injecting light into the Fiber 3 core. Calculated mode crosstalk matrices of pairs of the PLs with taper lengths of (**b**) 1.95 cm and (**c**) 2.05 cm, or with wavelengths at (**e**) 1530 nm and (**f**) 1620 nm.

## 4. Conclusions

In conclusion, we analyze the periodic evolution of the PL output modes with respect to taper lengths, affected by possible mode phase differences varying along the taper. Then, we design a three-mode OAM-MSPL for MDM with mode crosstalk below −24 dB and IL below −0.03 dB. As the simulations performed revealed, the designed OAM-MSPL has a taper length error tolerance longer than 1 mm with mode crosstalk below −21 dB and IL below −0.06 dB, and works stably in the whole C-band and L-band with low loss and crosstalk.

**Author Contributions:** This paper was built on original ideas by Y.L. (Yan Li) and Y.L. (Yang Li), and both of them performed the calculations. All authors contributed to the writing and editing of the manuscript.

**Funding:** This research was funded by the National Natural Science Foundation of China (NSFC) (61875019, 61675034, 61875020, 61571067), The Fund of State Key Laboratory of IPOC (BUPT), and The Fundamental Research Funds for the Central Universities.

**Conflicts of Interest:** The authors declare no conflict of interest.

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
