# Peer review of "Mode-Selective Photonic Lanterns for Orbital Angular Momentum Mode Division Multiplexing"

_applsci, doi:10.3390/app9112233_

Round 1
Reviewer 1 Report
Please see the attachment.

Author Response
Dear Reviewer,
Thanks very much for your kind letter, along with the constructive comments concerning our manuscript APPLSCI-486295, "Mode-Selective Photonic Lanterns for Orbital Angular Momentum Mode".
We have thoroughly considered all the comments of you and substantially revised our manuscript, and the major revised portions are marked highlight in our revised manuscript. We also respond point by point to your comments as listed below, along with a clear indication of the location of the revision. We look forward to hearing from you.
Sincerely yours,
Yang Li
Please see the attachment.

Reviewer 2 Report
I have revised the manuscript, and in my opinion, this paper could be considered to be published after addressing a few concerns:
English should be improved.
In Fig.1, why Fiber 1 is larger than Fiber 2 and 3? As I understand, during the tapering process, the claddings of three optical fibers become new cores, and surrounding microcapillary becomes a new cladding. I am not sure about keeping the dimensions of cores during the tapering process. Maybe it would be better if the photonic lantern was made of solid rods with desired refractive indices (and dimensions) surrounded by the capillary. Such a method is used for example in manufacturing photonic crystal fibers.
I think equations (1) and (2) should be rewritten. The authors should use superscript and subscript or at least use exp() rather than e^().
I have not noticed any experimental results confirming theoretical calculations. Do the authors consider conducting experiments towards the proposed design of photonic lanterns?
The list of references consists of 10 publications. I think the authors could extend it. Moreover, the comparison with results from other groups is highly welcomed.
I think Figure 5 should be placed near line 132. The same for Fig 4 - it should be near line 114. The authors should place figures below the text referring to them.
Author Response

(The authors gave the same response as above.)
